# Diet in the Prevention of Alzheimer’s Disease: Current Knowledge and Future Research Requirements

**DOI:** 10.3390/nu14214564

**Published:** 2022-10-30

**Authors:** Oliwia Stefaniak, Małgorzata Dobrzyńska, Sławomira Drzymała-Czyż, Juliusz Przysławski

**Affiliations:** Department of Bromatology, Poznan University of Medical Science, Rokietnicka 3 Street, 60-806 Poznan, Poland

**Keywords:** Alzheimer’s disease, diet, nutrients, DASH, Mediterranean diet, MIND diet

## Abstract

Alzheimer’s disease is a progressive brain disease that is becoming a major health problem in today’s world due to the aging population. Despite it being widely known that diet has a significant impact on the prevention and progression of Alzheimer’s disease, the literature data are still scarce and controversial. The application of the principles of rational nutrition for the elderly is suggested for Alzheimer’s disease. The diet should be rich in neuroprotective nutrients, i.e., antioxidants, B vitamins, and polyunsaturated fatty acids. Some studies suggest that diets such as the Mediterranean diet, the DASH (Dietary Approaches to Stop Hypertension) diet, and the MIND (Mediterranean-DASH Intervention for Neurodegenerative Delay) diet have a beneficial effect on the risk of developing Alzheimer’s disease.

## 1. Introduction

Along with the aging population, the number of cases of dementia is increasing. Alzheimer’s disease is the most common cause of dementia, 60–80% of cases. Alzheimer’s disease is a chronic illness, a major public health problem worldwide, and one of the most severe diseases affecting the brain. It affects cognitive function by worsening memory and can lead to people becoming unable to function independently in society. The etiology of Alzheimer’s disease is still not fully understood, and pharmacological treatment focuses only on relieving symptoms. According to current knowledge, it is believed that lifestyle factors may influence the risk of developing Alzheimer’s disease. These factors are, e.g., physical activity, smoking, alcohol consumption, and diet [1,2].

Despite common knowledge that diet has an impact on the risk of developing Alzheimer’s disease, literature data are still scarce and arouse a lot of controversy. Studies often discuss food groups, or a specific product and the nutrients that they contain. Their role is to assess which food elements have neuroprotective properties and may potentially impact the risk of developing Alzheimer’s disease. In this aspect antioxidants, B vitamins, and polyunsaturated fatty acids (PUFA) arouse the most significant interest.

Few studies analyze the general assumptions of the diet and the distribution of macronutrients in the diet. Most studies suggest that nutritional recommendations for the elderly should be used for people with Alzheimer’s disease, since Alzheimer’s disease generally affects people over 65 years old [2,3]. Some research suggests that using specific diets, such as the Mediterranean diet, the DASH (Dietary Approaches to Stop Hypertension) diet, and the MIND diet (Mediterranean-DASH Intervention for Neurodegenerative Delay diet), may help to protect from Alzheimer’s disease [2,4].

Nowadays, there is considerable research on the nutrition of people who have Alzheimer’s disease. However, these studies, analyzed separately, do not allow indisputable conclusions to be drawn, as comparisons of results in separate studies are a source of dispute.

The objective of this review was to evaluate studies on nutrition aimed at the prevention of Alzheimer’s disease. The research was conducted using PubMed, Scopus, Cochrane Library, Web of Sciences, and Embase. The study was identified with the use of the following phrases: “Alzheimer’s disease” and “diet”, and “Alzheimer’s disease” and “nutrients”. Additionally, the retrieved references were screened manually to find the relevant potential literature. A review of English language articles published until 6 June 2022 was retrieved.

## 2. Characteristics and Epidemiology of Alzheimer’s Disease

Alzheimer’s disease (AD) is the main cause of dementia in the elderly population. It is predicted that this number will increase due to the aging of society. Alzheimer’s Disease International (ADI) warns that the number of people who have Alzheimer’s disease may reach 13.8 million by 2060 [5].

Alzheimer’s disease is an incurable and progressive degenerative disease of the brain. It appears most often in people over 65 years of age, but it can also occur in youth. Two forms of this disease are distinguished:Early-onset Alzheimer’s disease (EOAD), which includes the familial form of familial Alzheimer’s disease (FAD) with a genetic basis,Late-onset Alzheimer’s disease (LOAD), where etiology is not fully understood.

The onset of Alzheimer’s disease is often asymptomatic. It is estimated that the symptoms of the disease can occur only about 20 years after the appearance of the first changes in the structure of the brain. These progressive changes, especially in the hippocampal area, lead to severe deficits of cognitive function and thus to the gradual dependence of the sick person on third parties [6,7].

In addition, this disease can also lead to brain atrophy, which is caused by a significant loss of brain cells and reduced brain capacity to metabolize glucose [8,9].

### 2.1. Factors Affecting the Development of Alzheimer’s Disease

Alzheimer’s disease is a disease of unknown etiology, which means that the factors conducive to its development are not well known. In the form of familial Alzheimer’s disease, genetics play a significant role [10].

Other unmodified factors correlating with the onset of Alzheimer’s disease are female sex and old age. It is observed that women are more likely to be morbid, and Alzheimer’s disease most often affects people over 65 years of age. In addition, a higher risk of developing Alzheimer’s disease is also associated with past brain injuries, i.e., concussion, post-traumatic amnesia, diabetes, obesity and hypertension in middle age, and a positive family history of Alzheimer’s disease [11].

However, it is worth paying special attention to modifiable factors that may reduce the risk of developing Alzheimer’s disease. These are level of education, physical activity, sleep, diet, smoking, or alcohol consumption. All these factors have a greater or lesser impact on the risk of developing cardiovascular disease, which also promotes the onset of Alzheimer’s disease. Diseases of the heart and blood vessels can cause reduced blood flow to the brain, thereby reducing the availability of nutrients. Scientific research indicates that regular physical activity, activities aimed at reducing the risk of obesity, diabetes, and cardiovascular diseases, as well as a healthy diet, i.e., the Mediterranean diet or the MIND diet, may have a beneficial effect on human cognitive resources and consequently contribute to reducing the risk of developing Alzheimer’s disease (Figure 1) [12].

Studies show that some factors that make up a person’s lifestyle impact the risk of developing and subsequent development of Alzheimer’s disease. Such factors may be, e.g., physical activity, and a healthy diet, which in the future may have a preventive effect against Alzheimer’s disease [13,14].

### 2.2. Potential Dietary Factors Leading to the Development of Alzheimer’s Disease

The influence of dietary factors in the development of Alzheimer’s disease has been suggested by some hypotheses. These factors include both a deficiency and an excess of dietary compounds. Deficiency of antioxidants in the diet, i.e., vitamins E and C, as well as folates, vitamins B_6_ and B_12_, could be a factor in the development of Alzheimer’s disease [15,16,17]. Antioxidant vitamins reduce β-amyloid-induced lipid peroxidation and oxidative stress and suppress inflammation signaling cascades [18,19]. Folic acid, vitamin B_6_, and vitamin B_12_ are involved in DNA methylation and are essential cofactors for homocysteine metabolism. Their deficiency may contribute to Alzheimer’s disease through increased homocysteine levels and subsequent oxidative damage [16,20].

A high fat diet and excess saturated fatty acids (SFA) may also underlie the development of Alzheimer’s disease. A higher intake of fats and SFAs is related to hyper-insulinemia, which is associated with a higher risk of Alzheimer’s disease [18,21]. Moreover, a high fat diet rich in SFAs promotes the development of hypercholesterolemia [22].

It has been indicated that there is a relationship between abnormalities in cholesterol metabolism and Alzheimer’s disease [23]. High cholesterol levels contribute to oxysterols accumulating in the brain in Alzheimer’s patients [24,25]. A study by Cutler et al. showed a positive correlation between cholesterol levels in the brain and the severity of dementia in Alzheimer’s disease patients [26].

In animal model studies, a high fat and high cholesterol diet may induce Tau hyperphosphorylation, negatively affect memory performance, and increase hippocampal p-tau levels in old age [27,28]. However, other studies do not indicate the association between dietary cholesterol and Alzheimer’s disease [29]. Therefore, further research is required.

There is speculation about the role of heavy metals (especially lead and cadmium) in the development Alzheimer’s disease, although current results are very inconclusive [30,31].

## 3. Nutrition in the Prevention and Alzheimer’s Disease

Literature data on diet in Alzheimer’s disease are scarce. They focus on the beneficial or harmful effects of individual food ingredients, but do not discuss the total amount of macro and micronutrient supply in this disease entity. Available data suggest that Alzheimer’s disease should follow the principles of rational nutrition for the elderly, which considers changes in the gastrointestinal tract of older people and the risk of individual deficiencies as a result [3]. In addition, studies indicate that the Mediterranean diet should form the basis of nutritional recommendations for people with Alzheimer’s disease, because of its beneficial effect upon their nutritional status and cognitive function [32].

### 3.1. Energy Demand

Moderate to severe cognitive impairment correlates positively with an increased risk of energy malnutrition and deficiencies in minerals and vitamins. Such cases include people who have Alzheimer’s disease. The onset of malnutrition in those people is a common phenomenon and contributes to the acceleration of disease progression [33,34]. Malnutrition, in their case, is affected, not only by age-related changes, but also by the disease itself. Studies show that changes in the central nervous system (CNS) in people with Alzheimer’s disease also apply to centers responsible for controlling appetite and regulating food consumption [35,36].

It has also been proven that these people are more likely to experience a decrease in taste function, contributing to a reduction in the number of meals consumed [33]. An important aspect in their case is also a change in eating habits. It is observed that people with Alzheimer’s disease often have problems with preparing and even eating meals on their own. Moreover, it happens that such people completely forget about the need to eat a meal, and due to the loss of appetite and regulation of meal intake, they may not take food or fluids for a prolonged period, which will lead to malnutrition [37].

Studies show that the living conditions of people with cognitive impairment, and more specifically whether they live in nursing homes or alone in their own homes, do not affect the risk of malnutrition, which allows us to presume that malnutrition in their case is associated with their disorders [38]. However, it is worth mentioning that both energy deficiency and energy surplus negatively affect the human body. People with Alzheimer’s disease may also risk taking in too many calories. Due to frontal lobe damage, they may be more likely to have episodes of binge eating.

On the other hand, excessive eating may also be caused by appetite disorders, or the fact that the patient forgets about the previously consumed meal and eats it again [39]. Excessive energy content of the diet in people who have Alzheimer’s disease may contribute to the occurrence of obesity. Studies show that obesity is associated with cognitive impairment and dementia and is as dangerous as malnutrition [40].

Incorrect nutritional status in people suffering from diseases in which there is a chronic state of dementia can increase the symptoms of the disease, e.g., sleep disorders, hallucinations, or apathy. In addition, such a condition is associated with a deterioration in functionality, which may result in the need to place people with such disorders in nursing homes. Therefore, a fundamental task is to detect potential abnormalities in the diet of people with the diseases mentioned above and provide them with a balanced diet [41,42].

There is a lot of controversy regarding the value of resting energy expenditure in people with Alzheimer’s disease. This accounts for about 60–75% of the total daily energy expenditure. There are hypotheses about hypermetabolism in people with Alzheimer’s disease [43]. Niskanen L. et al.’s previous research suggested that resting energy expenditure in people with Alzheimer’s disease is not significantly different from healthy people of the same age [44]. Other studies on the total daily energy expenditure of people with Alzheimer’s disease have even suggested reducing it by 14% compared to healthy older people [45].

However, the NUDAD (Nutrition the Unrecognized Determinant in Alzheimer’s Disease) project, carried out in recent years, shows the existence of hypermetabolism again. In a study conducted by Doorduijn A. S. et al. in patients with dementia during Alzheimer’s disease, or with mild cognitive impairment, it was noted that patients in the study group showed higher resting energy expenditure, despite the absence of significant differences in the total energy intake in both groups. This observation suggests that the increased metabolism—not the reduced energy supply—is associated with malnutrition in Alzheimer’s disease. These data may be useful in the future to optimize dietary recommendations for patients with dementia during Alzheimer’s disease [46].

In the case of Alzheimer’s disease, it is recommended to follow the nutrition guidelines for the elderly. According to current nutritional standards for the European population, it is estimated that people over the age of 65 years should take around 1500–2000 kcal daily [47]. However, older people are often unable to cover such energy demand, as evidenced by the incidence of malnutrition in them. In particular, attention should be paid to people who have Alzheimer’s disease, and, in whose case, malnutrition leads to a decrease in their functionality and worsens the disease’s symptoms [48].

### 3.2. Protein Supply in the Diet

The need for protein changes with age, and it depends to a large extent on the body’s physiological state, physical activity, or diseases [49]. In the literature, there are few works dealing strictly with the supply of protein in the diet of people who have Alzheimer’s disease. Current recommendations, however, draw attention to the importance of changing the amount of protein supply with age. Researchers suggest that an adequate supply of protein throughout life may act prophylactically against Alzheimer’s disease [49,50].

In a study conducted by Yeh T.S. et al., which in its assumption assessed the effect of long-term protein supply in the diet on cognitive function, it was noted that replacing 5% of energy in the diet from protein, with an equivalent amount of energy from carbohydrates, was associated with a higher probability of cognitive decline. In addition, when 5% of energy from vegetable protein was replaced with animal protein, a reduction in the risk of cognitive decline was observed. This study also suggested that consumption of legumes, fish, and lean meat has more beneficial effects on cognitive function than eating fast-food animal products. These data point out that, in the case of a protein supply, attention should be paid to its quantity and its quality [51].

As previously mentioned, it is suggested that the best solution in the case of people who have Alzheimer’s disease, also in the case of a protein supply, is a diet guided by the principles of rational nutrition in the elderly [3]. According to current nutritional standards, the percentage of protein in the diet in the elderly should be increased and it should around 20% of the daily energy requirement [52]. Such a supply of protein is recommended because, in old age, protein loss from the body is increased. The reason for this condition is for example chronic diseases, which, with age, occur more and more often and contribute to increasing the catabolism process in the body, thus increasing protein loss. An additional reason is a change in mass body composition, i.e., a decrease in lean body mass, which may lead to the occurrence of symptoms of sarcopenia, which in old age is a common phenomenon and is caused by reduced protein supply [53,54].

### 3.3. Supply of Fats in the Diet

As with protein, few papers in the scientific literature strictly address the amount of fat supply in the diet of people with Alzheimer’s disease. Studies conducted on animals assessed the consequences of a high-fat diet with Alzheimer’s disease. They were mostly focused on the effects of, e.g., type two diabetes, atherosclerosis and obesity caused by a high-fat diet on parameters such as blood–brain barrier permeability, β-amyloid aggregation, or insulin sensitivity. However, the results of these studies show many controversies, e.g., regarding the beneficial or harmful effects of a high-fat diet on Alzheimer’s disease [55,56,57].

In a study conducted by Rollins C. P. E. et al. on animal models of Alzheimer’s disease, the effect of obesity, caused by a high-fat diet during adolescence on Alzheimer’s disease, was observed. This study suggests that obesity in mice during adolescence, induced by a high-fat diet, may be associated with increased brain atrophy and memory impairment, and can consequently contribute to Alzheimer’s disease [58]. Another study suggesting that a high-fat diet may have an adverse effect on Alzheimer’s disease is the work of Wang M. et al., in which it has been observed that a high-fat diet promotes the aggregation of β-amyloid in the brain, which is one of the main features of Alzheimer’s disease [57]. The increase in β-amyloid aggregation in the brain caused by a high-fat diet was also shown by Mazzei G. et al. In this study, a mouse model that accumulated plaque β-amyloid from 6 months of age was used and showed mild cognitive impairment at 18 months of age. He indicated that the use of a high-fat diet in such mice contributed to obesity and impaired glucose tolerance, accompanied by a clear increase in β-amyloid aggregation [55].

However, it is worth mentioning that not all researchers suggest the adverse effect of a high-fat diet on Alzheimer’s disease. Elhaik Goldman S. et al. showed that mice given a high-fat diet had increased body weight, insulin resistance, and higher cholesterol levels. However, they did not show increased aggregation of β-amyloid. Additionally, in this study, mice on a control diet showed increased blood–brain barrier permeability and brain atrophy compared to mice on a high-fat diet, suggesting that the high-fat diet protects against blood–brain barrier disruption and brain atrophy in the studied mice. Researchers suggest that, in Alzheimer’s disease, a high-fat diet may benefit cognitive function by improving the functionality of the blood–brain barrier [59].

In Alzheimer’s disease, the amount of fat in the human diet is not the only important aspect. Its quality is equally important, particularly from which fatty acids it is made. It is suggested that unsaturated fatty acids have a protective effect against Alzheimer’s disease, while the consumption of SFA has a higher risk of developing the disease [60]. These data are confirmed by, e.g., AIDE (Cardiovascular risk factors, aging, and dementia) population studies. They showed that a high supply of total fat and SFA in the form of milk and spreads was associated with an increased risk of developing mild cognitive impairment. In addition, they linked frequent consumption of fish, rich in omega-3 unsaturated acids, with better cognitive function and better sensory memory [61]. 

The adverse effects of SFAs on Alzheimer’s disease are also suggested by Hill M. et al. This study concludes that dietary patterns that cause increased intake of saturated fatty acids have the effect on short- and long-term memory recall. This observation supports the earlier assumption that consuming a lot of SFAs is positively correlated with dementia and Alzheimer’s disease [62]. A ketogenic diet based on medium-chain triglycerides (MCTs) has also recently become the subject of scientists’ research. Medium-chain triglycerides in the human body are metabolized to ketone bodies and then they can function as an energy substrate for the brain [63]. 

One of the studies dealing with the impact of the diet mentioned above on Alzheimer’s disease was conducted by Ota M. et al. This study was conducted on people with Alzheimer’s disease but was limited by the small sample size. It found a beneficial effect of the ketogenic diet on improving working memory, short-term memory, and processing speed in patients suffering from mild or moderate Alzheimer’s disease [64]. These results are consistent with a study by Fortier M. et al., which demonstrated the effects of a ketogenic drink containing MCTs on cognitive function in people with mild cognitive impairment. This study, despite the high rate of resignation associated with adverse reactions, suggests that the consumption of medium-chain triglycerides improves cognitive function in mild cognitive impairment [65].

Current nutritional recommendations regarding the supply of fat in people with Alzheimer’s disease, as in the case of proteins, are the guidelines for rational nutrition for the elderly. According to the WHO, daily energy taken from fat should not exceed 30%. In addition, it is recommended that saturated fatty acids—SFAs should not exceed 10% of the total energy supply, and the daily supply of omega-3 and omega-6 unsaturated fatty acids should be about 250 mg [66,67].

### 3.4. Supply of Carbohydrates in the Diet

Studies suggest that glucose metabolism is associated with the development of Alzheimer’s disease and that people with impaired glycemia are more likely to develop dementia, and progress more rapidly from mild cognitive impairment to Alzheimer’s disease [68]. The effect of glucose metabolism on Alzheimer’s disease also confirms that people with type II diabetes and elevated blood glucose levels have a higher risk of developing Alzheimer’s disease than people with normoglycemia and without diabetes [69,70].

The need for carbohydrates in people with Alzheimer’s disease is not precisely defined in the same way as in the case of other macronutrients. In this case, attention should be paid to studies evaluating the impact on Alzheimer’s disease of diets with different carbohydrate content. One such study by Taylor M. K. et al., assessed the effects of a high-glycemic diet on the aggregation of β-amyloid and the cognitive function of healthy older adults. In this study, it was observed that 26% of the participants had an increase in the cerebral load of β-amyloid, and it was positively correlated with carbohydrate intake. Additionally, it was noted that sugar consumption was associated with poorer cognitive performance. These observations suggest that a high-glycemic diet may stimulate the aggregation of β-amyloid and lead to an increased risk of developing Alzheimer’s disease in the future [71]. 

The previously cited study conducted by Yeh T. S. et al., assessing the impact on cognitive function of long-term protein supply in the diet, is also worth mentioning. It suggests that converting the energy supplied in the form of protein into energy derived from carbohydrates may have a negative impact on human cognitive function [51]. This hypothesis is also confirmed by a study conducted by Shang X. et al., which aimed to investigate the relationship between the composition of macronutrients in breakfast and cognitive function. This study noted that replacing 5% of energy from carbohydrates with an equivalent amount of energy from protein or fat in breakfast was positively correlated with a lower rate of cognitive decline in older adults [72]. 

Summarizing the studies mentioned above, a high-carbohydrate diet, associated with increased blood glucose levels, seems to affect human cognitive function negatively [68,69,70,71,72]. However, not all carbohydrates have a negative impact on a person’s cognitive performance. Dietary fiber may be an interesting subject of study for Alzheimer’s disease, in view of the possible reduction in the risk of developing type II diabetes, which has been linked to an increased risk of developing Alzheimer’s disease [69,70,73]. In an animal study conducted by Shi H. et al., it was proven that deficiency of dietary fiber could lead to cognitive disorders, manifested by deficits in the memory of the location of objects, memory of the temporal order, and the ability to perform daily life activities. In addition to cognitive impairment, this study also noted that dietary fiber deficiency led to structural changes in the hippocampus and disturbances in the gut microbiota of mice, leading to subsequent cognitive impairment [74].

According to current recommendations on the nutrition of the elderly, the supply of carbohydrates should be 45–65% of the daily energy requirement. In addition, it is recommended that the energy from simple sugars does not exceed more than 10% of the daily energy requirement and adequate intake for dietary fiber should be about 20 g [75].

## 4. Nutrients with Neuroprotective Effect

The influence of nutritional factors on Alzheimer’s disease is related not only to macronutrients, but it has also been suggested that the main role in the prevention and treatment of the disease is played by factors with neuroprotective and antioxidant properties. The beneficial effect of nutritional factors on Alzheimer’s disease is shown in Figure 2.

### 4.1. Vitamins with Antioxidant Properties

Oxidative stress contributes to endogenous oxidative damage to proteins, lipids, and even DNA structure, which is considered one of the etiological factors of the aging process and of the development of chronic diseases, i.e., Alzheimer’s disease [76]. In the case of Alzheimer’s disease, oxidative stress is induced by β-amyloid, and its aggregation is associated with the formation of senile plaques. These plaques are deposited in the walls of blood vessels near the brain’s neurons, resulting, among other things, in the death of nerve cells [34].

Several studies on antioxidants suggest that they may have a beneficial effect on human cognitive function. Vitamins E and C have been at the center of the researchers’ attention. Despite the extensive amount of research on the effects of these two vitamins on Alzheimer’s disease, this topic is still controversial. According to the current literature review, there are studies that both confirm and refute the beneficial effects of these vitamins. The study proving the beneficial effects of vitamins E and C on human cognitive function is the population-based, prospective Rotterdam Study. This study lasted 6 years, and it was observed that a high dietary intake of vitamins C and E is associated with a reduced risk of developing Alzheimer’s disease [77]. 

A reduced risk of developing Alzheimer’s disease due to increased intake of vitamins C and E was suggested in a study by Zandi P. et al. [78]. The beneficial effects of vitamin E were demonstrated by Dysken M. et al. in their double-blind, placebo-controlled, randomized trial. They showed that supplementation of α-tocopherol at a dose of 2000 IU per day in people suffering from mild and moderate Alzheimer’s disease might slow the progression of the disease [79]. The opposite results were observed in the work of Liu H. et al. This study looked at assessing the causal relationship between vitamin C and Alzheimer’s disease. The results of this study did not show a significant link between plasma vitamin C levels and Alzheimer’s disease in people of European descent. However, they suggested that the relationship between vitamin C levels and the risk of developing Alzheimer’s may be gender-dependent [80].

In preventing and treating Alzheimer’s disease, another antioxidant deserves attention: vitamin D. Dursun E. et al. showed that serum vitamin D levels in Alzheimer’s patients are lower than in healthy people. It is worth noting that vitamin D deficiency is now considered a risk factor for Alzheimer’s disease for patients who are not carriers of ApoEε4 [81]. However, studies conducted to establish a link between vitamin D supplementation and a reduced risk of developing Alzheimer’s disease do not show its beneficial effects on cognitive or emotional functioning. Therefore, further studies are needed to evaluate the impact of vitamin D on cognitive function [82].

Although the beneficial effects of antioxidants as a group on human cognitive function are not conclusive, patients with Alzheimer’s disease are advised to have their diets rich in nutrients containing these compounds. Their daily diet should be rich in vegetables, fruits, and cold-pressed vegetable oils, the consumption of which has a positive effect on reducing the risk of developing cognitive disorders and dementia [83,84].

### 4.2. B Vitamins and Cognitive Function

B vitamins, particularly, are involved in DNA methylation and homocysteine metabolism. Elevated plasma homocysteine levels are a strong etiological factor in the risk of developing vascular dementia and Alzheimer’s disease. This leads to cognitive decline, white matter damage, brain atrophy, and dementia. It has been shown that people who have Alzheimer’s disease show elevated homocysteine levels. That is why, in these people, it is so important to maintain an appropriate concentration of B vitamins in the serum [85].

Homocysteine is also associated with increased susceptibility of neurons to oxidative stress related to Alzheimer’s disease [86]. That is why its proper metabolism becomes such an important issue, for which it is responsible, among other things, for the adequate concentration of B vitamins in the serum [85]. Literature data suggest that the consumption of B vitamins affects human cognitive function. The EMCOA (Effects and Mechanism Investigation of Cholesterol and Oxysterol on Alzheimer’s disease) study showed a relationship between vitamin B_12_ deficiency and cognitive decline. Additionally, it has been shown that the correct concentration of vitamins B_6_, B_9_, and B_12_ has a positive effect on the mental resources of healthy people [16]. The Smith A. et al. study showed that supplementation with vitamin B (B_6_, B_12_, and B_9_) could slow down the rate of accelerated brain atrophy that occurs in mild cognitive impairment [85].

### 4.3. Calcium and Magnesium and Their Role in Cognitive Impairment

There is a hypothesis that deterioration of calcium metabolism in neurons can lead to Alzheimer’s disease and brain aging through dendrite pruning, synaptic loss, aggregation of β-amyloid, tau protein, p-tau, inflammation, mitochondrial dysfunction, and oxidative stress. Thus, the improvement of calcium activity in neurons seems to be a promising strategy in actions aimed at reducing the risk of developing cognitive decline and Alzheimer’s disease [87].

The fact that magnesium is a natural physiological calcium channel blocker, initiated research into the effects of the Ca:Mg ratio on cognition. A PPCCT (Personalized Prevention of Colorectal Cancer Trial) study showed that reducing the Ca:Mg ratio by a personalized magnesium supplement over 12 weeks (diets with a high Ca:Mg ratio) significantly improved the overall MoCA test score, compared to the placebo group among participants over the age of 65. However, this supplementation did not affect the MoCA results in people under 65 [88].

### 4.4. Polyunsaturated Fatty Acids

Consumption of PUFAs, especially the omega-3 fatty acids DHA and EPA, is well known to have a beneficial effect on the regulation of inflammation, which occurs in people who have Alzheimer’s disease [89]. This influence was assessed in the WHICAP (Washington Heights-Hamilton Heights-Inwood Columbia Aging Project) study. It found that a lower risk of developing Alzheimer’s disease was associated with higher consumption of all long-chain unsaturated fatty acids and omega-3s. In addition, it also associated a higher risk of developing Alzheimer’s disease with a high intake of monounsaturated fatty acids and an increased total fat content [90]. 

The potential benefits of PUFAs, especially EPA and DHA acids, in preventing cognitive decline are becoming of clinical interest, also because of their proven neuroprotective properties, such as increasing neuroplasticity of nerve membranes, promoting synaptogenesis, modulating signal transduction pathways in neuronal cells, and alleviating the inflammatory state [91].

The Shinto L. et al. study showed that the combined use of omega-3 and α-lipoic acid supplementation for 12 months was associated with a slowing of both cognitive and functional decline in participants with mild to moderate Alzheimer’s disease. However, this study was carried out on a small group [92]. A study by Stavrinou P. et al. investigated the effect of using fatty acids in combination with other nutrients. It was a 6-month randomized, double-blind, placebo-controlled study. It examined the effect of supplementation with high-dose omega-3 and omega-6 acids, in combination with antioxidant vitamins, including vitamins A and E, on the cognitive function and functional abilities of elderly people with mild cognitive impairment (MCI). The results of this study showed that such supplementation may slow down the process of cognitive and functional deterioration in elderly people with mild cognitive impairment [93].

On the other hand, Quinn et al. showed that supplementing 2 g/day of DHA for 18 months did not slow the cognitive and functional decline rate in patients with mild to moderate Alzheimer’s disease [94]. Freund-Levi Y. et al. also found that supplementation of DHA and EPA for 6 months did not improve cognitive functions. However, some positive effects were observed in patients with very mild Alzheimer’s disease [95]. In Multidomain Alzheimer Preventive Trial, the 3-year omega-3 PUFA (800 mg DHA and 225 mg EPA) supplementation with or without multidomain intervention (physical activity, cognitive training, and nutritional advice) in elderly adults with memory complaints did not confirm the influence of supplementation on cognitive function in elderly adults with memory complaints [96]. Therefore, further research should assess the benefits of PUFA supplementation on cognitive decline specifically in individuals who are deficient for these nutrients [96] and at a stage preceding development of cognitive impairment.

### 4.5. Polyphenolic Compounds

Recent reports suggest that polyphenolic compounds, including flavonoids, phenolic acids, and tannins, may delay or prevent age-related cognitive decline [97]. The sources of these compounds are food of plant origin, including grapes, blueberries, olive oil, nuts, cocoa, green tea, and grapes [98,99,100].

Dark grapes are characterized by a rich content of phenolic acids, flavonoids, and stilbenes [98]. One study confirming the beneficial effects of polyphenols from grapes on human cognition is a study by Lamport D. et al. This study assessed the impact of 12 weeks of daily consumption of grape juice on cognition, driving performance, and blood pressure. This study improved direct spatial memory and driving performance compared to the control group [99]. These observations are consistent with the study by Haskell-Ramsay C. F. et al., which showed that purple grape juice could significantly improve cognitive aspects and mood [101].

The study by Lamport D. et al. has shown that flavonoid-rich grape juice significantly increases regional blood perfusion in the brain two hours after its consumption. Additionally, grape juice has a positive effect on the maintenance of protection and improves the health of the brain’s circulatory system [99]. The study by Sabogal M. et al. observed that quercetin reverses the histopathological features of Alzheimer’s disease and alleviates cognitive and emotional impairment in mice with the triple transgenic Alzheimer’s disease model [102]. Additionally, of note is the prospective environmental study conducted as part of the Rush Memory and Aging Project. This study assessed the association of flavanol consumption with accidental Alzheimer’s dementia and was conducted on a group of 921 participants. This study showed that higher consumption of flavanol from food sources, particularly kaempferol and isorhamnetin, may protect against the development of Alzheimer’s dementia [97].

Epigallocatechin-3-Gallate (EGCG), a polyphenol found in green tea extract, may be beneficial regarding Alzheimer’s disease prevention and treatment. Studies have shown that this compound has protective effects against neuronal damage and anti-inflammatory and antiatherogenic properties [103,104]. In vitro and in vivo (on animal model) studies have shown that EGCG induces a reduction in β-amyloid accumulation by modulating several biological mechanisms [105,106,107,108,109]. In a study by Nan S. et al. It was demonstrated that EGCG may diminish the hyperphosphorylation of the Tau protein, and downregulate Beta-Secretase 1 and β-amyloid 1–42 expression to improve the antioxidant system and learning and memory function of rats with Alzheimer disease [110]. However, EGCG dose levels and administration frequency require further research.

### 4.6. Other Nutrients with Anti-Inflammatory Effects

Methylxanthines are a purine-derived group of pharmacologic agents which are obtained from secondary plant metabolism and have a clinical application due to their Methylxanthines are a purine-derived group of pharmacologic agents, which are obtained from secondary plant metabolism and have a clinical application due to their stimulating effect [111]. They are present in the daily diet in popular products such as coffee, tea, energy drinks, and chocolate. Besides the well-established bronchoprotective effects, these compounds are also known to have anti-inflammatory and anti-oxidative properties and neuroprotective effects [112].

The most studied methylxanthine in preventing Alzheimer’s disease is caffeine, a widely consumed active substance in the western world. Some experimental studies in animal models have indicated that caffeine may suppress brain β-amyloid production, prevent memory impairment, cause microglial activation, and reduce hippocampal pro-inflammatory cytokines [113,114,115,116].

Navarro A. et al. showed an inverse linear association between total coffee consumption and the risk of all-cause mortality in the Mediterranean cohort. This association was stronger for people aged ≥55 years (HR: 0.67; 95% Cl: 0.52, 0.86) than for younger people, who showed no significant association with coffee consumption [117]. Moreover, Paz-Graniel I. et al. indicated total dietary caffeine intake was associated with better cognitive functioning in a Mediterranean cohort of elderly participants (aged 55–75 years) with metabolic syndrome [118]. However, long-term interventional studies are needed to clarify these associations and establish the mechanisms of influence and dose of caffeine required.

## 5. Diets Which Have a Beneficial Effect on Human Cognitive Function

### 5.1. The Mediterranean Diet

The Mediterranean diet may reduce the risk of developing both mild cognitive impairment and Alzheimer’s disease. It is characterized by high consumption of vegetables, fruits, nuts, legumes, unrefined grains and low consumption of meat and dairy products. In addition, it is worth noting that in the Mediterranean diet, the total fat content may be moderate or high, ranging from 30–40% of the total daily energy requirement. However, it characterizes a beneficial relationship between unsaturated fatty acids and SFAs, which relates to high consumption of olive oil and fish and low meat consumption. In this diet, observe also the increased consumption of red wine with meals, which is a valuable source of polyphenols [119].

The beneficial effects of the Mediterranean diet are attributed to the effects of food products and their nutrients with potential neuroprotective effects. These products include fish and nuts containing omega-3 fatty acids, wine containing polyphenols, and fruits, vegetables, and grains containing antioxidants [120,121].

The current literature data mostly confirm the beneficial effect of the Mediterranean diet on the incidence of cognitive disorders. The Anastasiou C. et al. study was conducted by 1865 participants aged 64 and over. During this study, 90 participants developed dementia, 68 were diagnosed with Alzheimer’s disease, and 223 developed mild cognitive impairment. Compliance with the Mediterranean diet was assessed using the Meal Frequency Questionnaire, and participants’ scores for adherence to the Mediterranean diet ranged from 0 to 55 units. This study linked higher scores on participants’ diet adherence to the Mediterranean diet with better scores in the cognitive areas of memory, language, executive function, and visual-spatial perception. It was also observed that people with dementia consumed less vegetables, fruits, and fish. It is worth mentioning here that consuming one serving of fish per week was associated with reducing the risk of developing dementia by as much as 9.8%. It has been shown that following the Mediterranean diet is associated with a reduced risk of developing dementia, suggesting a positive relationship between the Mediterranean diet and human cognitive abilities [122]. These results are in line with the EPIC-Spain cohort study with dementia of 16,160 healthy participants. The follow-up in this study was approximately 21.6 years, and 459 participants developed dementia during their follow-up. This observation also suggested that the Mediterranean diet may be associated with a reduced risk of developing dementia [123].

According to WHO, the Mediterranean diet can be recommended for adults with normal cognitive function and for people with mild cognitive impairment to reduce the risk of developing cognitive decline and dementia. In the case of Alzheimer’s disease, more research is still needed to assess the impact of the Mediterranean diet on this disease entity [124].

### 5.2. DASH Diet

The DASH diet is intended to prevent high blood pressure. Over the years, it has also become a subject of interest to scientists who study cognitive function. Interest in the DASH diet stems from the proven relationship between blood pressure and the proper functioning of cognitive function. Some studies suggest that hypertension and hypotension affect cerebral perfusion and, consequently, different cognitive domains [125,126].

The DASH diet and the Mediterranean diet are currently recommended to people suffering from hypertension, as one of the elements of blood pressure control. It is based on product groups such as fruit, vegetables, and low-fat dairy products. It also includes whole grains, lean meats, fish, nuts, and legumes [4]. Compared to the Mediterranean diet, the DASH diet reduces fat intake, especially SFAs, and dietary cholesterol. The consumption of red meat, sweets, and sugar-containing beverages is also restricted. This diet is rich in nutrients such as dietary fiber, calcium, magnesium, and potassium, with a low sodium intake of 2300 mg per day [125,127].

Current data from observational studies suggest that the DASH diet may positively affect cognitive function, including Alzheimer’s disease. The Cache County Study on Memory, Health, and Aging found a favorable association between the DASH diet and cognitive function. The results of this study showed that both the DASH diet and the Mediterranean diet were positively correlated with improved cognitive function. In addition, the study found that better Mini-Mental State Examination scores were associated with increased consumption of legumes, nuts, and whole grains [127]. These results are consistent with another prospective cohort study of the elderly by Tangney C. C. et al., where the DASH diet was associated with a slower rate of cognitive decline [128].

Research suggests the DASH diet may be associated with a lower risk of developing Alzheimer’s disease. One such study is a study by Morris M. C. et al. which suggested that strict adherence to the DASH diet and the Mediterranean diet and moderate adherence to the MIND diet may be associated with a lower risk of developing Alzheimer’s disease [129]. A study by Blumenthal J.A. et al. looked at the effect of lifestyle on the neurocognitive functioning of older people with vascular risk factors and cognitive impairment, without dementia. This study linked improved verbal memory outcomes to the DASH diet. Daily physical activity with greater aerobic capacity has also been found to improve verbal memory scores and contribute to better executive functioning and processing speed. This may suggest that combining the DASH diet with exercise may provide more cognitive benefits than diet alone. This observation allows the conclusion that a healthy lifestyle may contribute to the reduction of the risk of cognitive deterioration [130].

Despite the large amount of research confirming the beneficial effect of the DASH diet on cognitive function, there are also studies showing that it does not affect the human cognitive domain. One such study is a cohort study by Nishi S. K. et al. The cognitive analysis in this study lasted 2 years and showed that the DASH diet was not associated with better cognitive outcomes [131].

### 5.3. MIND Diet

The MIND diet is a combination of the Mediterranean diet and the DASH diet. It was created to reduce the risk of developing cognitive disorders, including Alzheimer’s disease. The MIND diet, as well as the Mediterranean diet and the DASH diet, are based on plant-based foods. It places great emphasis on consuming large amounts of leafy greens, nuts, and blueberries, due to their neuroprotective effects [132].

The MIND diet is based on ten groups of products that might have beneficial effects on brain health and five groups of products that are contraindicated. Recommended groups of products include green leafy vegetables, other vegetables, nuts, berries, legumes, whole grain products, fish, poultry, olive oil, and red wine [132].

Green leafy vegetables contain potentially neuroprotective substances, such as folate, lutein, kaempferol, and β-carotene. The recommended intake in the MIND diet should be over six portions per week. The study by Morris M. C. et al. found that eating around one portion of leafy greens per day can slow cognitive decline. The guidelines in the MIND diet also recommend eating at least one vegetable, other than green leafy vegetables, daily and to eat legumes at least three times a week [133].

Blueberries are a rich source of anthocyanins. Recommended intake in this diet is at least twice a week. Carey A. N. et al. have confirmed that the effects of blueberries on brain function are an animal model. This study suggests that eating blueberries may be associated with lower rates of microglia activation and improved neuroplasticity, which may be associated with better memory function in the future [134]. It is also worth noting that the MIND diet allows the consumption of red wine in one portion a day (10 g of ethanol). Grape wine is a rich source of polyphenols, including resveratrol, which also has potential neuroprotective properties [135].

The main fat sources in the MIND diet are nuts and olive oil. It is recommended to eat nuts frequently, including in the form of a snack, and their supply should be at least five portions a week (1 portion approx. 30 g). Nuts are a rich source of linolenic acid, phytosterols, vitamin B1, and vitamin E. The literature has proven that eating nuts is associated with a lower risk of developing type II diabetes [129,136,137].

The source of protein in the MIND diet is fish and poultry. According to this diet, poultry should be eaten at least twice a week and should not be fried. Fish is a rich source of PUFAs, including DHA and EPA. The recommended fish consumption in this diet should be greater than one serving of fish per week. The study by Sanchez-Romero L. et al. analyzed people who had Alzheimer’s disease and proved that the consumption of fish oil rich in DHA and EPA is associated with a better condition of the erythrocyte membrane and a decrease in oxidative stress. Additionally, in the MIND diet, whole grains are a source of vegetable protein, and their recommended consumption is at least three servings of whole grains per day [129,132,138].

Food contraindicated in the MIND diet is red meat, butter, cheese, sweets, fried products, and fast food, which is high in sugar and saturated fatty acids. Some studies have found that sugar consumption correlates with poorer cognitive performance and eating lean meat or fish, rather than eating junk food, performs better in cognitive domains. In the MIND diet, it is recommended to eat red meat no more than four times a week, sweets no more than five times a week, and fast food and cheese no more than once a week. Additionally, it is recommended that the supply of butter and margarine should not exceed more than one tablespoon per day [129].

Studies assessing the impact of the MIND diet on human cognition and Alzheimer’s disease suggest that the MIND diet may be positively correlated with a reduced risk of cognitive impairment. A study by Morris M.C. et al. associated the MIND diet with a slowing decline in general cognitive function and individual cognitive domains, i.e., episodic memory, semantic memory, working memory, speed of perception, and organization of perception [139]. A study by Cherian L. et al., that assessed the effects of the MIND diet on cognitive function in stroke survivors, showed that the MIND diet slowed down the decline in overall cognition and semantic memory [140].

A cross-sectional, population-based study by McEvoy C. T. et al. supports better cognitive function caused by the MIND diet. This study indicates that the MIND diet is associated with better cognitive function and reduces the risk of cognitive impairment [141]. A 12-year-old Australian study conducted by Hosking D. E. et al. [142] also suggested a lower risk of cognitive impairment associated with the MIND diet. The direct effect of the MIND diet on the risk of developing Alzheimer’s disease was assessed in a study by Morris M. C. et al. The follow-up in this study lasted approximately 4.5 years and showed that the use of the MIND diet was associated with a reduced risk of developing Alzheimer’s disease, and that the MIND diet may provide greater cognitive benefits than the Mediterranean diet and the DASH diet [129].

The MIND diet is a beneficial dietary pattern for Alzheimer’s disease. It influences a slowing of the decline in cognitive function in general, and individual domains of cognition [129,139,140,141,142]. The contraindicated and recommended products, as well as the main aspects of the described diets for the prevention of Alzheimer’s disease, are presented in Table 1.

## 6. Alzheimer’s Disease Prevention—Main Guidelines

Based on a literature review, a proper balanced diet, rich in antioxidative and anti-inflammatory nutrients, may have a beneficial effect on the prevention of Alzheimer disease [84,89,93]. Additionally, the diet should avoid highly processed foods, rich in saturated and trans fatty acids, and poor quality food, which may increase exposure to the consumption of pollution and toxins, such as heavy metals [60]. The comprehensive list of nutritional factors for preventing Alzheimer’s disease is presented in Table 2.

## 7. Conclusions

Diet, at each stage of life, impacts the risk of developing Alzheimer’s disease. Literature data suggest that people with Alzheimer’s disease should follow the nutritional recommendations for the elderly. Additionally, the diet should be rich in neuroprotective substances, i.e., antioxidants, B vitamins, and polyunsaturated fatty acids. The Mediterranean diet, the DASH diet, and the MIND diet reduce the risk of developing Alzheimer’s disease.

## Figures and Tables

**Figure 1 nutrients-14-04564-f001:**
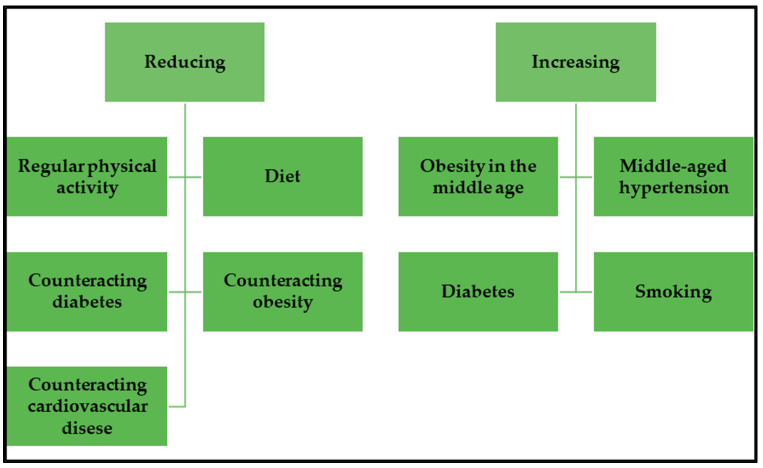
Factors influence the risk of developing Alzheimer’s disease.

**Figure 2 nutrients-14-04564-f002:**
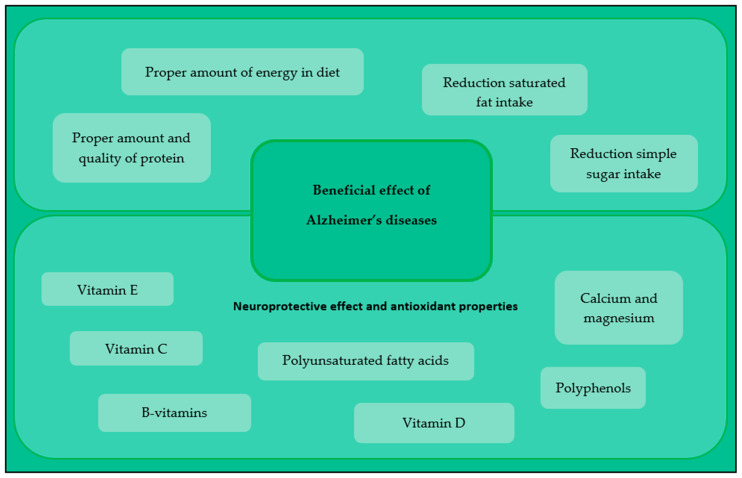
The effect of nutritional factors on Alzheimer’s disease.

**Table 1 nutrients-14-04564-t001:** Comparison of Mediterranean, DASH, and MIND diet—major aspect, recommended and contraindicated food products.

	The Mediterranean Diet	DASH Diet	MIND Diet
**Major aspect**	The plant-based diet consists of minimally processed products based on the traditional diet of individuals living in the Mediterranean region.	The plant-based diet consists of all minimally processed products used for cardiometabolic conditions especially reduced hypertension.	It is a combination of the Mediterranean diet and the DASH diet. It strictly defines the types of ten recommended and five contraindicated products.
		It focuses on reducing the intake of saturated fatty acids, trans fats, and sodium in the diet.	The plant-based diet consists of all minimally processed products used to improve brain cognition and reduce the risk of certain age-related neurodegenerative diseases.
**Recommended**	vegetables, fruits nuts legumes whole grains extra virgin olive oil olives fish and seafood moderate consumption of fermented dairy red wine in limited quantities	vegetables fruits nuts legumes whole grain products low-fat dairy products	green leafy vegetables (≥6 p/week) other vegetables (≥1 p/div.) nuts (≥5 p/week) berries (≥ 2 p/week) pods (≥3 p/week) whole grains (≥3 p/div.) poultry (≥2 p/week) fish and seafood (≥1 p/week) olive oil (basic oil) wine (1 p/div.)
	* portions are not exactly specified	* portions are not exactly specified	
**Contraindicated**	red meat sweets and sugars	total fat intake red meat salt and sodium-rich products sweets and sugars	red meat (<4 p/week) butter and margarine (<1 tbsp/div.) cheese (<1 p/week) cakes and sweets (<5 p/week) fried and fast-food products (<1 p/week)

**Table 2 nutrients-14-04564-t002:** The influence of nutritional factors on the prevention of Alzheimer’s disease.

Positive Dietary Factors	Negative Dietary Factors
Foods rich in antioxidative and anti-inflammatory compounds:	Highly processed products- fast food, ready-to-eat meals Saturated fatty acid (e.g., animal fats, palm oil, highly processed food) Trans fatty acids (e.g., partially hydrogenated fats in foods, meat) Simple sugar (e.g., sweets and sugar) Poor quality food—risk of excess consumption of pollution and toxins, e.g., heavy metal
Vitamin E (e.g., cold pressed vegetable oils) Vitamin C (e.g., citrus fruits, berries, acerola, peppers, broccoli) B-vitamins (e.g., nuts, seeds, beans, whole grain product) Polyunsaturated fatty acid (e.g., fish and seafood, nuts, camelina oil, linseed oil) Polyphenols (e.g., grapes, berry fruit, green tee, red wine)	

## Data Availability

No new data were created or analyzed in this study. Data sharing is not applicable to this article.

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
