# Peer review of "Diet in the Prevention of Alzheimer’s Disease: Current Knowledge and Future Research Requirements"

_nutrients, 2022, doi:10.3390/nu14214564_

Round 1
Reviewer 1 Report
The Review article entitled "Diet in the prevention of Alzheimer's disease: current knowledge and future research requierements" is a well written review and covers multiple upto date literature that handles diet strategies which aim to prevent the progression or occurance of Alzheimer's diesease.
Therefore the manuscript fits in the scope of the journal.
The authors categorize each metabolic group e.g. “energy”, “protein”, “fat” and give subsequent examples with citations of current literature articles. Especially good is that, the authors leave each category with a recommendation from the WHO or another study, which tells us how much amount of each macro/micronutrient we should absorb.
Another strength of this review is, that the authors not only include studys with a positive outcome, but also negative outcome.
Major:
However in my opinion. If we talk about nutrition and alzheimer’s disease some micronutrients are missing on the authors list:
For e.g. EGCG (https://onlinelibrary.wiley.com/doi/10.1002/mnfr.201000641) (https://infectagentscancer.biomedcentral.com/articles/10.1186/s13027-017-0145-6)
and
Methylxanthines (https://www.mdpi.com/1420-3049/21/8/974) .
Both act as antioxidative and anti-inflammatory chemicals that interfere with the pathomechanism of AD.
Especially coffee with one of its major component “caffeine” being a methylxanthine is highly involved in the global diet and also takes place in the mediterranean diet. (https://doi.org/10.1093/ajcn/nqy198)
For me this was a major missing point in this review and I think these should be included or the authors should go more into these, if already preexisting.
Minor:
I think the conclusion would be greatly improved if the authors would create a small table, where they could compare each diet in short terms of the major aspects and how they differ from each other, or what they have in common.
All in all this is a great review which leads the research field more in the direction of prevention through nutrition, which is the first part every human being can change by himself, if one is proper educated.
Author Response
Response to the comments made by the reviewers
Manuscript ID: nutrients-1902634
Title: Diet in the prevention of Alzheimer's disease: current knowledge and future research requirements
We would like to thank the Reviewers for their careful review of our manuscript and for providing us with some suggestions to improve its quality. We have carried out a major revision of the manuscript, and we believe the paper has improved significantly.
According to the Reviewers' suggestion, the manuscript has been carefully checked and corrected. The changes in the manuscript have been highlighted in red.
Below we sequentially address all of the points raised by the Reviewers.
Reviewer 1:
Firstly, we would like to express our profound thanks to the Reviewer for devoting time to reviewing our manuscript, the corrections and suggestions. We have carried out a major revision of the manuscript, and we believe the paper has improved significantly.
Major comments:
The Reviewer's comment: However in my opinion. If we talk about nutrition and alzheimer’s disease some micronutrients are missing on the authors list:
For e.g. EGCG (https://onlinelibrary.wiley.com/doi/10.1002/mnfr.201000641) (https://infectagentscancer.biomedcentral.com/articles/10.1186/s13027-017-0145-6)
and
Methylxanthines (https://www.mdpi.com/1420-3049/21/8/974) .
Both act as antioxidative and anti-inflammatory chemicals that interfere with the pathomechanism of AD.
Especially coffee with one of its major component “caffeine” being a methylxanthine is highly involved in the global diet and also takes place in the mediterranean diet. (https://doi.org/10.1093/ajcn/nqy198)
For me this was a major missing point in this review and I think these should be included or the authors should go more into these, if already preexisting.
The authors' answer: According to the Reviewer's suggestion, the changes have been made in the manuscript.
Epigallocatechin-3-Gallate (EGCG), a polyphenol found in green tea extract, may be beneficial regarding Alzheimer's disease prevention and treatment. Studies have shown that this compound has protective effects against neuronal damage and anti-inflammatory and antiatherogenic properties [83,84]. In vitro and in vivo (on animal model) studies have shown that EGCG induces a reduction in β-amyloid accumulation by modulating several biological mechanisms [85–89]. In a study by Nan S. et al. It was demonstrated that EGCG may diminish the hyperphosphorylation of the Tau protein, and downregulate Beta-Secretase 1 and β-amyloid 1-42 expression to improve the antioxidant system and learning and memory function of rats with Alzheimer disease [90]. However, EGCG dose levels and administration frequency require further re-search.
4.6. Other nutrients with anti-inflammatory effects
Methylxanthines are a purine-derived group of pharmacologic agents which are obtained from secondary plant metabolism and have a clinical application due to their Methylxanthines are a purine-derived group of pharmacologic agents, which are obtained from secondary plant metabolism and have a clinical application due to their stimulating effect [91]. They are present in the daily diet in popular products such as coffee, tea, energy drinks, and chocolate. Besides the well-established bronchoprotective effects, these compounds are also known to have anti-inflammatory and anti-oxidative properties and neuroprotective effects [92].
The most studied methylxanthine in preventing Alzheimer's disease is caffeine, a widely consumed active substance in the western world. Some experimental studies in animal models have indicated that caffeine may suppress brain β-amyloid production, prevent memory impairment, cause microglial activation, and reduce hippocampal pro-inflammatory cytokines [93–96].
Navarro A. et al. showed an inverse linear association between total coffee consumption and the risk of all-cause mortality in the Mediterranean cohort. This association was stronger for people aged ≥55 years (HR: 0.67; 95% Cl: 0.52, 0.86) than for younger people, who showed no significant association with coffee consumption [97]. Moreover, Paz-Graniel I. et al. indicated total dietary caffeine intake was associated with better cognitive functioning in a Mediterranean cohort of elderly participants (aged 55–75 years) with metabolic syndrome [98]. However, long-term, interventional studies are needed to clarify these associations and establish the mechanisms of influence and dose of caffeine required.
Minor comments:
The Reviewer's comment: I think the conclusion would be greatly improved if the authors would create a small table, where they could compare each diet in short terms of the major aspects and how they differ from each other, or what they have in common.
The authors' answer: According to the Reviewer's suggestion, we have added the Table 1 to manuscript which comparing the Mediterranean diet, DASH and MIND diets.
- Cascella, M.; Bimonte, S.; Muzio, M.R.; Schiavone, V.; Cuomo, A. The Efficacy of Epigallocatechin-3-Gallate (Green Tea) in the Treatment of Alzheimer’s Disease: An Overview of Pre-Clinical Studies and Translational Perspectives in Clinical Practice. Infect Agents Cancer 2017, 12, 36, doi:10.1186/s13027-017-0145-6.
- Forester, S.C.; Lambert, J.D. The Role of Antioxidant versus Pro-Oxidant Effects of Green Tea Polyphenols in Cancer Prevention. Mol. Nutr. Food Res. 2011, 55, 844–854, doi:10.1002/mnfr.201000641.
- He, M.; Liu, M.-Y.; Wang, S.; Tang, Q.-S.; Yao, W.-F.; Zhao, H.-S.; Wei, M.-J. [Research on EGCG improving the degenerative changes of the brain in AD model mice induced with chemical drugs]. Zhong Yao Cai 2012, 35, 1641–1644.
- Lin, C.-L.; Chen, T.-F.; Chiu, M.-J.; Way, T.-D.; Lin, J.-K. Epigallocatechin Gallate (EGCG) Suppresses Beta-Amyloid-Induced Neurotoxicity through Inhibiting c-Abl/FE65 Nuclear Translocation and GSK3 Beta Activation. Neurobiol Aging 2009, 30, 81–92, doi:10.1016/j.neurobiolaging.2007.05.012.
- Chang, X.; Rong, C.; Chen, Y.; Yang, C.; Hu, Q.; Mo, Y.; Zhang, C.; Gu, X.; Zhang, L.; He, W.; et al. (-)-Epigallocatechin-3-Gallate Attenuates Cognitive Deterioration in Alzheimer’s Disease Model Mice by Upregulating Neprilysin Expression. Exp Cell Res 2015, 334, 136–145, doi:10.1016/j.yexcr.2015.04.004.
- Lee, Y.-J.; Choi, D.-Y.; Yun, Y.-P.; Han, S.B.; Oh, K.-W.; Hong, J.T. Epigallocatechin-3-Gallate Prevents Systemic Inflammation-Induced Memory Deficiency and Amyloidogenesis via Its Anti-Neuroinflammatory Properties. J Nutr Biochem 2013, 24, 298–310, doi:10.1016/j.jnutbio.2012.06.011.
- Zhang, Z.-X.; Li, Y.-B.; Zhao, R.-P. Epigallocatechin Gallate Attenuates β-Amyloid Generation and Oxidative Stress Involvement of PPARγ in N2a/APP695 Cells. Neurochem Res 2017, 42, 468–480, doi:10.1007/s11064-016-2093-8.
- Nan, S.; Wang, P.; Zhang, Y.; Fan, J. Epigallocatechin-3-Gallate Provides Protection Against Alzheimer’s Disease-Induced Learning and Memory Impairments in Rats. Drug Des Devel Ther 2021, 15, 2013–2024, doi:10.2147/DDDT.S289473.
- Monteiro, J.P.; Alves, M.G.; Oliveira, P.F.; Silva, B.M. Structure-Bioactivity Relationships of Methylxanthines: Trying to Make Sense of All the Promises and the Drawbacks. Molecules 2016, 21, E974, doi:10.3390/molecules21080974.
- Janitschke, D.; Lauer, A.A.; Bachmann, C.M.; Grimm, H.S.; Hartmann, T.; Grimm, M.O.W. Methylxanthines and Neurodegenerative Diseases: An Update. Nutrients 2021, 13, 803, doi:10.3390/nu13030803.
- Larsson, S.; Orsini, N. Coffee Consumption and Risk of Dementia and Alzheimer’s Disease: A Dose-Response Meta-Analysis of Prospective Studies. Nutrients 2018, 10, 1501, doi:10.3390/nu10101501.
- Cao, C.; Cirrito, J.R.; Lin, X.; Wang, L.; Wang, L.; Verges, D.K.; Dickson, A.; Mamcarz, M.; Zhang, C.; Mori, T.; et al. Caffeine Suppresses Amyloid-Beta Levels in Plasma and Brain of Alzheimer’s Disease Transgenic Mice. J Alzheimers Dis 2009, 17, 681–697, doi:10.3233/JAD-2009-1071.
- Arendash, G.W.; Schleif, W.; Rezai-Zadeh, K.; Jackson, E.K.; Zacharia, L.C.; Cracchiolo, J.R.; Shippy, D.; Tan, J. Caffeine Protects Alzheimer’s Mice against Cognitive Impairment and Reduces Brain Beta-Amyloid Production. Neuroscience 2006, 142, 941–952, doi:10.1016/j.neuroscience.2006.07.021.
- Steger, R.; Kamal, A.; Lutchman, S.; Intrabartolo, L.; Sohail, R.; Brumberg, J.C. Chronic Caffeine Ingestion Causes Microglia Activation, but Not Proliferation in the Healthy Brain. Brain Res Bull 2014, 106, 39–46, doi:10.1016/j.brainresbull.2014.05.004.
- Navarro, A.M.; Martinez-Gonzalez, M.Á.; Gea, A.; Grosso, G.; Martín-Moreno, J.M.; Lopez-Garcia, E.; Martin-Calvo, N.; Toledo, E. Coffee Consumption and Total Mortality in a Mediterranean Prospective Cohort. The American Journal of Clinical Nutrition 2018, 108, 1113–1120, doi:10.1093/ajcn/nqy198.
- Paz-Graniel, I.; Babio, N.; Becerra-Tomás, N.; Toledo, E.; Camacho-Barcia, L.; Corella, D.; Castañer-Niño, O.; Romaguera, D.; Vioque, J.; Alonso-Gómez, Á.M.; et al. Association between Coffee Consumption and Total Dietary Caffeine Intake with Cognitive Functioning: Cross-Sectional Assessment in an Elderly Mediterranean Population. Eur J Nutr 2021, 60, 2381–2396, doi:10.1007/s00394-020-02415-w.

Reviewer 2 Report
The manuscript explains the effect of some types of diets on AD. It doesn't discuss how to prevent it using an appropriate diet! The authors didn't discuss the possible nutritional factors that may lead to AD. In that way, they could have suggested which foods to avoid in order not to get sick with AD.
Author Response
Response to the comments made by the reviewers
Manuscript ID: nutrients-1902634
Title: Diet in the prevention of Alzheimer's disease: current knowledge and future research requirements
We would like to thank the Reviewers for their careful review of our manuscript and for providing us with some suggestions to improve its quality. We have carried out a major revision of the manuscript, and we believe the paper has improved significantly.
According to the Reviewers' suggestion, the manuscript has been carefully checked and corrected. The changes in the manuscript have been highlighted in red.
Below we sequentially address all of the points raised by the Reviewers.
Reviewer 2:
Firstly, we would like to express our profound thanks to the Reviewer for devoting time to reviewing our manuscript, the corrections and suggestions. We have carried out a major revision of the manuscript, and we believe the paper has improved significantly.
The Reviewer's comments: The manuscript explains the effect of some types of diets on AD. It doesn't discuss how to prevent it using an appropriate diet! The authors didn't discuss the possible nutritional factors that may lead to AD. In that way, they could have suggested which foods to avoid in order not to get sick with AD.
The authors' answer: According to the Reviewer's suggestion, we have modified and completed the manuscript. We have added a summary chapter with nutritional factors for Alzheimer's disease prevention. Additionally, we have completed the information on methylxanthines and EGCG used in the prevention of Alzheimer's disease.
Based on a literature review, a proper balanced diet, rich in antioxidative and anti-inflammatory nutrients, may have a beneficial effect on the prevention of Alzheimer disease [67,73,75]. Additionally, the diet should avoid highly processed foods, rich in saturated and trans fatty acids, and poor quality food, which may increase exposure to the consumption of pollution and toxins, such as heavy metals [43]. The comprehensive list of nutritional factors for preventing Alzheimer's disease is presented in Table 2.
Table 2. The influence of nutritional factors on the prevention of Alzheimer's disease.
|
Positive dietary factors |
Negative dietary factors |
|
Foods rich in antioxidative and anti-inflammatory compounds: · Vitamin E (e.g. cold pressed vegetable oils) · Vitamin C (e.g. citrus fruits, berries, acerola, peppers, broccoli) · B-vitamins (e.g. nuts, seeds, beans, whole grain product) · Polyunsaturated fatty acid (e.g. fish and seafood, nuts, camelina oil, linseed oil) · Polyphenols (e.g. grapes, berry fruit, green tee, red wine) |
· Highly processed products- fast food, ready-to-eat meals · Saturated fatty acid (e.g. animal fats, palm oil, highly processed food) · Trans fatty acids (e.g. partially hydrogenated fats in foods, meat) · Simple sugar (e.g. sweets and sugar) · Poor quality food – risk of excess consumption of pollution and toxins e.g. heavy metal |
Epigallocatechin-3-Gallate (EGCG), a polyphenol found in green tea extract, may be beneficial regarding Alzheimer's disease prevention and treatment. Studies have shown that this compound has protective effects against neuronal damage and anti-inflammatory and antiatherogenic properties [83,84]. In vitro and in vivo (on animal model) studies have shown that EGCG induces a reduction in β-amyloid accumulation by modulating several biological mechanisms [85–89]. In a study by Nan S. et al. It was demonstrated that EGCG may diminish the hyperphosphorylation of the Tau protein, and downregulate Beta-Secretase 1 and β-amyloid 1-42 expression to improve the antioxidant system and learning and memory function of rats with Alzheimer disease [90]. However, EGCG dose levels and administration frequency require further re-search.
4.6. Other nutrients with anti-inflammatory effects
Methylxanthines are a purine-derived group of pharmacologic agents which are obtained from secondary plant metabolism and have a clinical application due to their Methylxanthines are a purine-derived group of pharmacologic agents, which are obtained from secondary plant metabolism and have a clinical application due to their stimulating effect [91]. They are present in the daily diet in popular products such as coffee, tea, energy drinks, and chocolate. Besides the well-established bronchoprotective effects, these compounds are also known to have anti-inflammatory and anti-oxidative properties and neuroprotective effects [92].
The most studied methylxanthine in preventing Alzheimer's disease is caffeine, a widely consumed active substance in the western world. Some experimental studies in animal models have indicated that caffeine may suppress brain β-amyloid production, prevent memory impairment, cause microglial activation, and reduce hippocampal pro-inflammatory cytokines [93–96].
Navarro A. et al. showed an inverse linear association between total coffee consumption and the risk of all-cause mortality in the Mediterranean cohort. This association was stronger for people aged ≥55 years (HR: 0.67; 95% Cl: 0.52, 0.86) than for younger people, who showed no significant association with coffee consumption [97]. Moreover, Paz-Graniel I. et al. indicated total dietary caffeine intake was associated with better cognitive functioning in a Mediterranean cohort of elderly participants (aged 55–75 years) with metabolic syndrome [98]. However, long-term, interventional studies are needed to clarify these associations and establish the mechanisms of influence and dose of caffeine required.
- Laitinen, M.H.; Ngandu, T.; Rovio, S.; Helkala, E.-L.; Uusitalo, U.; Viitanen, M.; Nissinen, A.; Tuomilehto, J.; Soininen, H.; Kivipelto, M. Fat Intake at Midlife and Risk of Dementia and Alzheimer’s Disease: A Population-Based Study. Dement Geriatr Cogn Disord 2006, 22, 99–107, doi:10.1159/000093478.
- McGrattan, A.M.; McGuinness, B.; McKinley, M.C.; Kee, F.; Passmore, P.; Woodside, J.V.; McEvoy, C.T. Diet and Inflammation in Cognitive Ageing and Alzheimer’s Disease. Curr Nutr Rep 2019, 8, 53–65, doi:10.1007/s13668-019-0271-4.
- Simonetto, M.; Infante, M.; Sacco, R.L.; Rundek, T.; Della-Morte, D. A Novel Anti-Inflammatory Role of Omega-3 PUFAs in Prevention and Treatment of Atherosclerosis and Vascular Cognitive Impairment and Dementia. Nutrients 2019, 11, 2279, doi:10.3390/nu11102279.
- S. Stavrinou, P.; Andreou, E.; Aphamis, G.; Pantzaris, M.; Ioannou, M.; S. Patrikios, I.; D. Giannaki, C. The Effects of a 6-Month High Dose Omega-3 and Omega-6 Polyunsaturated Fatty Acids and Antioxidant Vitamins Supplementation on Cognitive Function and Functional Capacity in Older Adults with Mild Cognitive Impairment. Nutrients 2020, 12, 325, doi:10.3390/nu12020325.
- Cascella, M.; Bimonte, S.; Muzio, M.R.; Schiavone, V.; Cuomo, A. The Efficacy of Epigallocatechin-3-Gallate (Green Tea) in the Treatment of Alzheimer’s Disease: An Overview of Pre-Clinical Studies and Translational Perspectives in Clinical Practice. Infect Agents Cancer 2017, 12, 36, doi:10.1186/s13027-017-0145-6.
- Forester, S.C.; Lambert, J.D. The Role of Antioxidant versus Pro-Oxidant Effects of Green Tea Polyphenols in Cancer Prevention. Mol. Nutr. Food Res. 2011, 55, 844–854, doi:10.1002/mnfr.201000641.
- He, M.; Liu, M.-Y.; Wang, S.; Tang, Q.-S.; Yao, W.-F.; Zhao, H.-S.; Wei, M.-J. [Research on EGCG improving the degenerative changes of the brain in AD model mice induced with chemical drugs]. Zhong Yao Cai 2012, 35, 1641–1644.
- Lin, C.-L.; Chen, T.-F.; Chiu, M.-J.; Way, T.-D.; Lin, J.-K. Epigallocatechin Gallate (EGCG) Suppresses Beta-Amyloid-Induced Neurotoxicity through Inhibiting c-Abl/FE65 Nuclear Translocation and GSK3 Beta Activation. Neurobiol Aging 2009, 30, 81–92, doi:10.1016/j.neurobiolaging.2007.05.012.
- Chang, X.; Rong, C.; Chen, Y.; Yang, C.; Hu, Q.; Mo, Y.; Zhang, C.; Gu, X.; Zhang, L.; He, W.; et al. (-)-Epigallocatechin-3-Gallate Attenuates Cognitive Deterioration in Alzheimer’s Disease Model Mice by Upregulating Neprilysin Expression. Exp Cell Res 2015, 334, 136–145, doi:10.1016/j.yexcr.2015.04.004.
- Lee, Y.-J.; Choi, D.-Y.; Yun, Y.-P.; Han, S.B.; Oh, K.-W.; Hong, J.T. Epigallocatechin-3-Gallate Prevents Systemic Inflammation-Induced Memory Deficiency and Amyloidogenesis via Its Anti-Neuroinflammatory Properties. J Nutr Biochem 2013, 24, 298–310, doi:10.1016/j.jnutbio.2012.06.011.
- Zhang, Z.-X.; Li, Y.-B.; Zhao, R.-P. Epigallocatechin Gallate Attenuates β-Amyloid Generation and Oxidative Stress Involvement of PPARγ in N2a/APP695 Cells. Neurochem Res 2017, 42, 468–480, doi:10.1007/s11064-016-2093-8.
- Nan, S.; Wang, P.; Zhang, Y.; Fan, J. Epigallocatechin-3-Gallate Provides Protection Against Alzheimer’s Disease-Induced Learning and Memory Impairments in Rats. Drug Des Devel Ther 2021, 15, 2013–2024, doi:10.2147/DDDT.S289473.
- Monteiro, J.P.; Alves, M.G.; Oliveira, P.F.; Silva, B.M. Structure-Bioactivity Relationships of Methylxanthines: Trying to Make Sense of All the Promises and the Drawbacks. Molecules 2016, 21, E974, doi:10.3390/molecules21080974.
- Janitschke, D.; Lauer, A.A.; Bachmann, C.M.; Grimm, H.S.; Hartmann, T.; Grimm, M.O.W. Methylxanthines and Neurodegenerative Diseases: An Update. Nutrients 2021, 13, 803, doi:10.3390/nu13030803.
- Larsson, S.; Orsini, N. Coffee Consumption and Risk of Dementia and Alzheimer’s Disease: A Dose-Response Meta-Analysis of Prospective Studies. Nutrients 2018, 10, 1501, doi:10.3390/nu10101501.
- Cao, C.; Cirrito, J.R.; Lin, X.; Wang, L.; Wang, L.; Verges, D.K.; Dickson, A.; Mamcarz, M.; Zhang, C.; Mori, T.; et al. Caffeine Suppresses Amyloid-Beta Levels in Plasma and Brain of Alzheimer’s Disease Transgenic Mice. J Alzheimers Dis 2009, 17, 681–697, doi:10.3233/JAD-2009-1071.
- Arendash, G.W.; Schleif, W.; Rezai-Zadeh, K.; Jackson, E.K.; Zacharia, L.C.; Cracchiolo, J.R.; Shippy, D.; Tan, J. Caffeine Protects Alzheimer’s Mice against Cognitive Impairment and Reduces Brain Beta-Amyloid Production. Neuroscience 2006, 142, 941–952, doi:10.1016/j.neuroscience.2006.07.021.
- Steger, R.; Kamal, A.; Lutchman, S.; Intrabartolo, L.; Sohail, R.; Brumberg, J.C. Chronic Caffeine Ingestion Causes Microglia Activation, but Not Proliferation in the Healthy Brain. Brain Res Bull 2014, 106, 39–46, doi:10.1016/j.brainresbull.2014.05.004.
- Navarro, A.M.; Martinez-Gonzalez, M.Á.; Gea, A.; Grosso, G.; Martín-Moreno, J.M.; Lopez-Garcia, E.; Martin-Calvo, N.; Toledo, E. Coffee Consumption and Total Mortality in a Mediterranean Prospective Cohort. The American Journal of Clinical Nutrition 2018, 108, 1113–1120, doi:10.1093/ajcn/nqy198.
- Paz-Graniel, I.; Babio, N.; Becerra-Tomás, N.; Toledo, E.; Camacho-Barcia, L.; Corella, D.; Castañer-Niño, O.; Romaguera, D.; Vioque, J.; Alonso-Gómez, Á.M.; et al. Association between Coffee Consumption and Total Dietary Caffeine Intake with Cognitive Functioning: Cross-Sectional Assessment in an Elderly Mediterranean Population. Eur J Nutr 2021, 60, 2381–2396, doi:10.1007/s00394-020-02415-w.

Round 2
Reviewer 1 Report
The authors adressed all my comments. Therefore it is now ready to be published.
Author Response
Response to the comments made by the reviewers
Manuscript ID: nutrients-1902634
Title: Diet in the prevention of Alzheimer's disease: current knowledge and future research requirements
We would like to express our profound thanks to the Reviewer for devoting time to reviewing our manuscript and give positive feedback.
Reviewer 2 Report
The manuscript is improved and some of my concerns are addressed, but they didn't reply to one of my concerns. The authors didn't discuss the possible nutritional factors that may lead to AD.
Author Response
Response to the comments made by the reviewers
Manuscript ID: nutrients-1902634
Title: Diet in the prevention of Alzheimer's disease: current knowledge and future research requirements
Dear Reviewer, we appreciate all your insightful comments. Thank you for your suggestions.
The changes in the manuscript have been highlighted in red. Below we sequentially address all of the points raised by the Reviewers.
The Reviewer's comments: The manuscript is improved and some of my concerns are addressed, but they didn't reply to one of my concerns. The authors didn't discuss the possible nutritional factors that may lead to AD.
The authors' answer: According to the Reviewer's suggestion, we have modified and completed the manuscript. We have added a paragraph with a potential dietary factors leading to the development Alzheimer's disease. Additionally, the work was checked by a native speaker.
2.2. Potential dietary factors leading to the development of Alzheimer's disease
The influence of dietary factors in the development of Alzheimer's disease has been suggested by some hypotheses. These factors include both a deficiency and an excess of dietary compounds. Deficiency of antioxidants in the diet, i.e., vitamins E and C, as well as folates, vitamins B6 and B12, could be a factor in the development of Alzheimer's disease [15–17]. Antioxidant vitamins reduce β-amyloid-induced lipid peroxidation and oxidative stress and suppress inflammation signaling cascades [18,19]. Folic acid, vitamin B6, and vitamin B12 are involved in DNA methylation and are essential cofactors for homocysteine metabolism. Their deficiency may contribute to Alzheimer's disease through increased homocysteine levels and subsequent oxidative dam-age [16,20].
A high fat diet and excess saturated fatty acids (SFA) may also underlie the development of Alzheimer's disease. A higher intake of fats and SFAs is related to hyper-insulinemia, which is associated with a higher risk of Alzheimer's disease [18,21]. Moreover, a high fat diet rich in SFAs promotes the development of hypercholesterolemia [22].
It has been indicated that there is a relationship between abnormalities in cholesterol metabolism and Alzheimer's disease [23]. High cholesterol levels contribute to oxysterols accumulating in the brain in Alzheimer's patients [24,25]. A study by Cutler et al. showed a positive correlation between cholesterol levels in the brain and the se-verity of dementia in Alzheimer's disease patients [26].
In animal model studies, a high fat and high cholesterol diet may induce Tau hyperphosphorylation, negatively affect memory performance, and increase hippocampal p-tau levels in old age [27,28]. However, other studies do not indicate the association between dietary cholesterol and Alzheimer's disease [29]. Therefore, further re-search is required.
There is speculation about the role of heavy metals (especially lead and cadmium) in the development Alzheimer's disease, although current results are very inconclusive [30,31].
- Mielech, A.; Puścion-Jakubik, A.; Markiewicz-Żukowska, R.; Socha, K. Vitamins in Alzheimer’s Disease—Review of the Latest Reports. Nutrients 2020, 12, 3458, doi:10.3390/nu12113458.
- An, Y.; Feng, L.; Zhang, X.; Wang, Y.; Wang, Y.; Tao, L.; Qin, Z.; Xiao, R. Dietary Intakes and Biomarker Patterns of Folate, Vitamin B6, and Vitamin B12 Can Be Associated with Cognitive Impairment by Hypermethylation of Redox-Related Genes NUDT15 and TXNRD1. Clin Epigenet 2019, 11, 139, doi:10.1186/s13148-019-0741-y.
- Mecocci, P.; Polidori, M.C. Antioxidant Clinical Trials in Mild Cognitive Impairment and Alzheimer’s Disease. Biochimica et Biophysica Acta (BBA) - Molecular Basis of Disease 2012, 1822, 631–638, doi:10.1016/j.bbadis.2011.10.006.
- Luchsinger, J.A.; Mayeux, R. Dietary Factors and Alzheimer’s Disease. The Lancet Neurology 2004, 3, 579–587, doi:10.1016/S1474-4422(04)00878-6.
- Allan Butterfield, D.; Castegna, A.; Drake, J.; Scapagnini, G.; Calabrese, V. Vitamin E and Neurodegenerative Disorders Associated with Oxidative Stress. Nutritional Neuroscience 2002, 5, 229–239, doi:10.1080/10284150290028954.
- Agnew-Blais, J.C.; Wassertheil-Smoller, S.; Kang, J.H.; Hogan, P.E.; Coker, L.H.; Snetselaar, L.G.; Smoller, J.W. Folate, Vitamin B-6, and Vitamin B-12 Intake and Mild Cognitive Impairment and Probable Dementia in the Women’s Health Initiative Memory Study. Journal of the Academy of Nutrition and Dietetics 2015, 115, 231–241, doi:10.1016/j.jand.2014.07.006.
- Luchsinger, J.A.; Tang, M.-X.; Shea, S.; Mayeux, R. Hyperinsulinemia and Risk of Alzheimer Disease. Neurology 2004, 63, 1187–1192, doi:10.1212/01.wnl.0000140292.04932.87.
- Maffeis, C.; Cendon, M.; Tomasselli, F.; Tommasi, M.; Bresadola, I.; Fornari, E.; Morandi, A.; Olivieri, F. Lipid and Saturated Fatty Acids Intake and Cardiovascular Risk Factors of Obese Children and Adolescents. Eur J Clin Nutr 2021, 75, 1109–1117, doi:10.1038/s41430-020-00822-0.
- Feringa, F.M.; van der Kant, R. Cholesterol and Alzheimer’s Disease; From Risk Genes to Pathological Effects. Front. Aging Neurosci. 2021, 13, 690372, doi:10.3389/fnagi.2021.690372.
- Poli, G.; Biasi, F.; Leonarduzzi, G. Oxysterols in the Pathogenesis of Major Chronic Diseases. Redox Biology 2013, 1, 125–130, doi:10.1016/j.redox.2012.12.001.
- Testa, G.; Staurenghi, E.; Zerbinati, C.; Gargiulo, S.; Iuliano, L.; Giaccone, G.; Fantò, F.; Poli, G.; Leonarduzzi, G.; Gamba, P. Changes in Brain Oxysterols at Different Stages of Alzheimer’s Disease: Their Involvement in Neuroinflammation. Redox Biology 2016, 10, 24–33, doi:10.1016/j.redox.2016.09.001.
- Cutler, R.G.; Kelly, J.; Storie, K.; Pedersen, W.A.; Tammara, A.; Hatanpaa, K.; Troncoso, J.C.; Mattson, M.P. Involvement of Oxidative Stress-Induced Abnormalities in Ceramide and Cholesterol Metabolism in Brain Aging and Alzheimer’s Disease. Proc Natl Acad Sci U S A 2004, 101, 2070–2075, doi:10.1073/pnas.0305799101.
- Glöckner, F.; Meske, V.; Lütjohann, D.; Ohm, T.G. Dietary Cholesterol and Its Effect on Tau Protein: A Study in Apolipoprotein E-Deficient and P301L Human Tau Mice. J Neuropathol Exp Neurol 2011, 70, 292–301, doi:10.1097/NEN.0b013e318212f185.
- Crisby, M.; Rahman, S.M.A.; Sylvén, C.; Winblad, B.; Schultzberg, M. Effects of High Cholesterol Diet on Gliosis in Apolipoprotein E Knockout Mice. Neuroscience Letters 2004, 369, 87–92, doi:10.1016/j.neulet.2004.05.057.
- Ylilauri, M.P.; Voutilainen, S.; Lönnroos, E.; Mursu, J.; Virtanen, H.E.; Koskinen, T.T.; Salonen, J.T.; Tuomainen, T.-P.; Virtanen, J.K. Association of Dietary Cholesterol and Egg Intakes with the Risk of Incident Dementia or Alzheimer Disease: The Kuopio Ischaemic Heart Disease Risk Factor Study. Am J Clin Nutr 2017, 105, 476–484, doi:10.3945/ajcn.116.146753.
- Bakulski, K.M.; Seo, Y.A.; Hickman, R.C.; Brandt, D.; Vadari, H.S.; Hu, H.; Park, S.K. Heavy Metals Exposure and Alzheimer’s Disease and Related Dementias. JAD 2020, 76, 1215–1242, doi:10.3233/JAD-200282.
- Lee, H.J.; Park, M.K.; Seo, Y.R. Pathogenic Mechanisms of Heavy Metal Induced-Alzheimer’s Disease. Toxicol. Environ. Health Sci. 2018, 10, 1–10, doi:10.1007/s13530-018-0340-x.
